# NEURAL NONMYOPIC BAYESIAN OPTIMIZATION IN DYNAMIC COST SETTINGS

**Sang T. Truong**[1], **Duc Q. Nguyen**[1,2], **Willie Neiswanger**[3], **Ryan-Rhys Griffiths**[4],
**Stefano Ermon**[1], **Nick Haber**[1], **Sanmi Koyejo**[1]
[1]Stanford, [2]HCMUT - VNU-HCM, [3]University of Southern California, [4]FutureHouse, Inc.

## ABSTRACT

Bayesian optimization (BO) is a popular framework for optimizing black-box functions, leveraging probabilistic models such as Gaussian processes. Conventional BO algorithms, however, assume static query costs, which limit their applicability to real-world problems with dynamic cost structures such as geological surveys or biological sequence design, where query costs vary based on the previous actions. We propose a novel nonmyopic BO algorithm named LookaHES featuring dynamic cost models to address this. LookaHES employs a neural network policy for variational optimization over multi-step lookahead horizons to enable planning under dynamic cost environments. Empirically, we benchmark LookaHES on synthetic functions exhibiting varied dynamic cost structures. We subsequently apply LookaHES to a real-world application in protein sequence design using a large language model policy, demonstrating its scalability and effectiveness in handling multi-step planning in a large and complex query space. LookaHES consistently outperforms its myopic counterparts in synthetic and real-world settings, significantly improving efficiency and solution quality. Our implementation is available at https://github.com/sangttruong/nonmyopia.

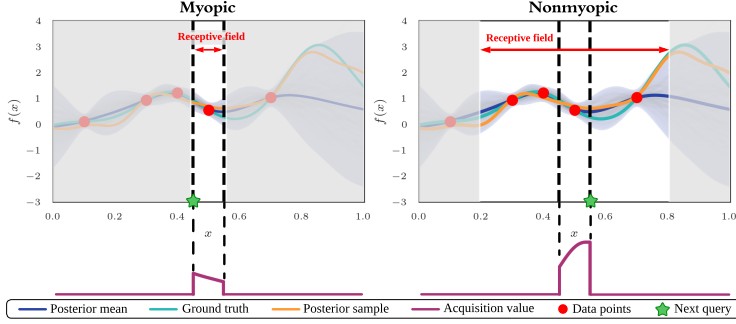

Figure 1: Comparison of myopic and nonmyopic Bayesian optimization in a dynamic cost environment. The receptive field, shown as the *unshaded* region, is defined as *the subset of the input space that the decision-maker (DM) considers when selecting the next query*. The myopic strategy (left) has a narrow receptive field, as the DM focuses only on short-term gains. In contrast, in the nonmyopic strategy (right), the receptive field expands by considering future queries, allowing the DM to "invest" - making queries that may initially seem suboptimal but unlock better opportunities in the future (see the acquisition value in the bottom row). Nonmyopic policy enables the DM to navigate the cost structure strategically, ultimately reaching high-value regions that would have been inaccessible with a purely myopic strategy. See Section 3.1 for a detailed discussion.

## 1 INTRODUCTION

Bayesian optimization (BO) (Kushner, 1962; 1964; Shahriari et al., 2016; Frazier, 2018; Garnett, 2022) is a powerful tool for optimizing black-box functions by employing a probabilistic surrogate model, typically a Gaussian process (GP), together with an acquisition function, to balance exploration and exploitation of the unknown objective function. In conventional BO, query costs are

typically assumed to be static. The assumption of static query costs can be an obstacle to applying BO in practical applications where query costs may vary dynamically on a per-iteration basis (Aglietti et al., 2021; Lee et al., 2021; Folch et al., 2022; 2024). For instance, in geological surveys, the cost of querying a location varies based on its proximity to the previous query due to transportation expenses (Bordas et al., 2020). Another example is biological sequence design, where editing one token at a time incurs a low cost, but moving beyond the edit distance of one token becomes prohibitively expensive (Guo et al., 2004; Błażej et al., 2017). These environments exhibit a dynamic cost structure, where the query cost at a given location might depend on the last query or even the entire query history. Incorporating these cost structures into the decision-making process can significantly improve the solution quality returned by BO algorithms. These cost structures dynamically constrain the effective input space where the decision-making algorithm can move, requiring the agent to plan its decision by looking at multiple steps in the future.

As cost structures become more complex and interdependent, myopic strategies may fail to capture long-term benefits, highlighting the need for nonmyopic BO. Nonmyopic BO incorporates lookahead steps to make more informed decisions at the current timestep (González et al., 2016; Astudillo et al., 2021; Yue & Kontar, 2020; Jiang et al., 2020a). One potential approach to solve nonmyopic BO in a dynamic cost environment is to view it as a Markov Decision Process (MDP) (Garcia & Rachelson, 2013; Puterman, 2014). MDPs are commonly used to model sequential decision-making problems. In this context, where we aim to determine the optimal next action in a sample-efficient manner, an MDP frames the decision process as a cost-constrained model-based reinforcement learning (CMBRL) problem, where the queried inputs are states and actions influence the transitions between consecutive states. Traditional CMBRL approaches, which rely on world models to simulate the environment (Janner et al., 2019; Wang & Ba, 2020; Hafner et al., 2021; Hamed et al., 2024), are not directly suitable to non-myopic BO settings.

On the one hand, off-the-shelf CMBRL methods are inadequate for meeting the diverse requirements of many nonmyopic BO applications in dynamic cost settings. Specifically, they often struggle with large, complex, and semantically rich action spaces, as they are typically integrated with simple neural network policies designed for small, discrete action spaces (Janner et al., 2019; Wang & Ba, 2020; Hafner et al., 2021). In the above example of biological sequences, using a simple model is often inadequate to incorporate domain knowledge during policy optimization. A policy with strong domain knowledge is critical when dealing with high-dimensional and complex action spaces, like editing sequences, where each action has rich semantic meaning and can significantly impact the outcomes (Stolze et al., 2015). Recent literature has demonstrated that using pre-trained Large Language Models (LLMs) that encode vast quantities of domain-specific knowledge as the policy offers an exciting approach to exploit the semantic structures in various real-world action spaces (Palo et al., 2023; Zhuang et al., 2024; Hazra et al., 2024). Unfortunately, existing RL frameworks designed to work with LLMs primarily focus on myopic policies in contextual bandit settings (Ouyang et al., 2024). Recent popular frameworks (von Werra et al., 2020; Hu et al., 2024; Zheng et al., 2024; Harper et al., 2019) mainly focus on techniques for single-turn reinforcement learning. Hence, these existing frameworks can not be directly applied in a nonmyopic BO setting. On the other hand, these methods are unnecessarily complex for various applications in nonmyopic BO. For example, in biological sequence design, a biologist edits specific amino acids in the initial sequence. These edits deterministically define state transitions, eliminating the need for a stochastic model.

Another limitation of CMBRL is its difficulty in managing reward uncertainties (Ez-zizi et al., 2023). In nonmyopic BO, handling uncertainty is essential (Treven et al., 2024; Sun et al., 2024) for effectively balancing exploitation and exploration (Zangirolami & Borrotti, 2024). Typically, CMBRL algorithms utilize neural reward models which tend to be poorly calibrated (Minderer et al., 2021; Zhao et al., 2024), resulting in overconfident or underconfident reward estimation and potentially leading to suboptimal actions (Sun et al., 2024). To mitigate the limits of exploration in CMBRL and hence the probability of selecting suboptimal actions, recent research emphasizes accounting for reward uncertainties rather than relying solely on average values (Lötjens et al., 2019; Luis et al., 2023; Ez-zizi et al., 2023). This approach enables the use of various acquisition functions to model aleatoric and epistemic uncertainty, allowing policies to better adapt to dynamic or noisy environments, such as biological sequence wet-lab testing, where even minor changes or errors can significantly alter the final results (Caraus et al., 2015).

To address these challenges, we propose a cost-constrained nonmyopic BO algorithm named Looka-HES, which reduces the exponential complexity of optimizing multiple decision variables while

maintaining strong exploration capabilities through a computationally efficient Bayesian reward model. Additionally, this method can be applied across diverse domains, from sequence design to natural language processing, where multiple interactions are required before a final decision is made. Our contributions are summarized as follows.

- We formulate the problem of nonmyopic BO in dynamic cost settings with various cost models inspired by real-world scenarios, such as ones in biological sequence design.
- We utilize a neural policy to variationally optimize decision variables for nonmyopic Bayesian optimization in dynamic cost settings. LookaHES demonstrates scalability to a lookahead horizon of at least 20 steps, significantly surpassing the state-of-the-art, which typically only extends to four steps within a similar computational budget.
- We benchmark LookaHES against baselines across nine synthetic functions ranging from 2D to 8D with varying noise levels and on a real-world problem involving NASA satellite images. Utilizing a recurrent neural network policy, LookaHES consistently outperforms traditional acquisition functions.
- We demonstrate the effectiveness of LookaHES by applying it to constrained protein sequence design. We also developed an open-source, scalable framework that enhances the efficiency of policy optimization for LLM-based policies within a complex dynamic cost environment.

## 2 BACKGROUND

### 2.1 NEURAL NONMYOPIC BAYESIAN OPTIMIZATION

The decision maker (DM) aims to find the maximum of a black-box function $f^* : \mathcal{X} \to \mathcal{Y}$ through a sequence of $T$ queries. Here, $\mathcal{Y}$ is a subset of $\mathbb{R}$. To do so, DM makes $T$ queries $x_{1:T} = [x_1, \ldots, x_T]$ and observes the corresponding outputs $y_{1:T} = [y_1, \ldots, y_T]$. The output $y_t$ is obtained by evaluating the query $x_t \in \mathcal{X}$, with noise modeled as $y_t = f^*(x_t) + \epsilon_t$, where $\epsilon_t$ represents the noise. Given a prior distribution over the parameters $p(\theta)$, a probabilistic surrogate model $f_\theta$ of the black-box function $f^*$ is sampled by $\theta \sim p(\theta)$. The posterior distribution of the function, conditioned on the history of queries and observations up to time step $t$, is given by $p_t(\theta) = p(\theta|D_t) = p(\theta|x_1, y_1, \ldots, x_t, y_t)$. In this work, we utilize Gaussian Process and Gaussian linear regression as our surrogate models. Details of these design choices are presented in Section 4. After $T$ queries, DM selects an action $a \in \mathcal{A}$. In general Bayesian decision-making literature, $\mathcal{A}$ can be distinct from $\mathcal{X}$. For example, the DM might query the black box function to find the top-k or level set. This paper focuses on global optimization; hence, the action set is the same as the query set, $\mathcal{A} = \mathcal{X}$[1].

At each time step $t$, the DM selects $x_t$ either directly or indirectly via a policy model, with the objective to maximize expected information gain (Neiswanger et al., 2022). The optimization objective typically comprises a loss function and, where applicable, a cost function. In this work, we primarily consider the loss function $\ell(f^*, a) = -f^*(a)$. Details of the optimization objective are presented in Section 3.

Following Russo & Van Roy (2016); Kandasamy et al. (2018), the Bayesian cumulative regret at timestep $T$ is defined as $\mathbb{E}\left[\sum_{t=1}^T \left(f^*(a^*) - f^*(a_t)\right)\right]$, where the expectation is taken over the randomness from the environment, the sequence of queries, and the final actions. Figure 3 illustrates the conceptual view of our optimization process. We summarize our notations in Appendix A.

### 2.2 COST STRUCTURES IN BAYESIAN OPTIMIZATION

We briefly review the related work on cost-sensitive BO by providing a taxonomy of the fields. We provide further details on other related aspects of the literature in Appendix B. We classify cost structures into four categories based on uncertainty (known or unknown) and variability (dynamic or static) (Table 1). When the cost is static, BO literature divides it into two variations: homogeneous cost, where all queries have the same cost, and heterogeneous cost, where the cost depends on the

---

[1]We still use the notion of a final action to be consistent with the literature. We note that our method can apply to a general action set, and studying that is beyond the scope of our paper.

Table 1: Classification of cost types based on uncertainty and variability.

| | **Known Cost** | **Unknown Cost** |
|---|---|---|
| **Static Cost** | Costs that remain constant regardless of the queries made during optimization. These costs are predictable and can be pre-determined, making them straightforward to budget and plan. **Related papers**: Wu & Frazier (2019); Nyikosa et al. (2018); Lam et al. (2016) | Costs that remain unaffected by the sequence of queries but whose exact amount is uncertain due to external factors (e.g., system fluctuations or resource availability). While they are static, their unpredictability complicates cost estimation. **Related papers**: Astudillo et al. (2021); Lee et al. (2021); Snoek et al. (2012); Luong et al. (2021) |
| **Dynamic Cost** | Costs change based on the sequence of previous queries. These costs are influenced by past optimization steps but remain predictable, allowing for some level of planning. **Related papers**: Liu et al. (2023), this paper | Costs depend on previous queries and are unpredictable, making them difficult to estimate in advance. These costs often arise in environments with high variability, such as dynamic resource allocation or uncertain execution times. **Related papers**: To our knowledge, no prior work exists. |

To illustrate the distinction between cost structures, we visualize the uncertainty and variability of these structures as probabilistic graphical diagrams (Figure 2). In this figure, $f$ represents the target black-box function, $x$ denotes the input query, $y$ is the output value, and $c$ is the cost of querying $x$. On the left — the dynamic-cost structure — the cost of querying $x_3$ can depend on $x_1$ and $x_2$. On the right — the static-known cost structure — the cost of querying $x_3$ is independent

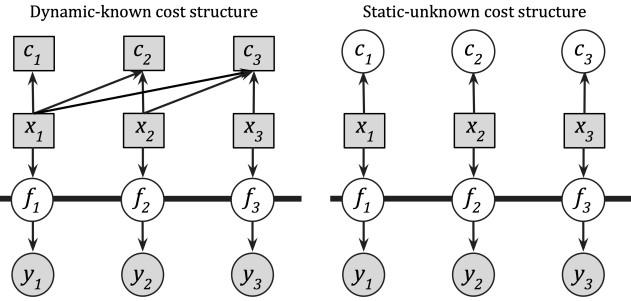

Figure 2: Graphical models of two popular cost structures.

Our study addresses Bayesian optimization within a known and dynamic cost setting, a structure distinct from the cost scenarios explored in prior works. For instance, while (Liu et al., 2023) proposes an Euclidean cost-constrained lookahead acquisition function similar to ours, it lacks pathwise sampling and a variational network, leading to exponential growth in computational requirements. By integrating these components, our LookaHES method significantly reduces complexity, enabling longer lookahead horizons and improved global optimization. Other works (Astudillo et al., 2021; Lee et al., 2021; Luong et al., 2021; Snoek et al., 2012) consider unknown, heterogeneous costs in settings like hyperparameter optimization, where costs are static for specific configurations. Meanwhile, other approaches in (Wu & Frazier, 2019; Nyikosa et al., 2018; Lam et al., 2016) do not account for cost structures, assuming constant, known costs across the search space. Nonmyopic methods from (Astudillo et al., 2021; Lee et al., 2021; Snoek et al., 2012; Wu & Frazier, 2019) address multi-step optimization using free variables, incurring exponential complexity as the lookahead horizon expands. In contrast, LookaHES incorporates a language model policy, further distinguishing it by offering a scalable approach to dynamic cost optimization.

## 2.3 DYNAMIC-KNOWN COST STRUCTURE

For scenarios where the costs of queries change dynamically, we define the cost of querying $x_t$ as $c(x_{<t}, x_t)$, where $c$ is an application-specific cost function provided to the decision-maker. The total cost to execute $T$ queries, $x_{1:T}$, is given by $\sum_{t=1}^{T} c(x_{<t}, x_t)$. We define two primary cost structures:

(i) Markovian cost, depending only on the previous query, and (ii) non-Markovian cost, depending on the entire query history. The Markovian cost is incurred based on the location of departure $x_{t-1}$ and the destination $x_t$. It also depends on the $p$-norm between $x_t$ and $x_{t-1}$. The relationship between distance and cost in practice can be nonlinear: for example, traveling within a ball of radius of $r$ might be free, but beyond that, the traveling cost grows at a rate of $k$. The observed cost might be perturbed by a random noise $\epsilon$. These ideas are summarized in the following cost model: $c_{\text{Markov}}(x_{t-1}, x_t) = \max(k(||x_t - x_{t-1}||_p - r), 0) + \epsilon$. Euclidean cost ($p = 2, r = 0$), Manhattan cost ($p = 1, r = 0$), and $r$-spotlight cost ($k = \infty$) are some commonly used instances. Euclidean cost is found in applications such as ground surveys since the traveling cost depends on the distance between departure and arrival locations (Bordas et al., 2020). Spotlight cost is found in biological sequence design, where editing more than one token is impossible in one experiment (Belanger et al., 2019). Regarding non-Markovian costs, the query cost could depend on the entire query history. For example, the traveler in the ground survey application might participate in a mileage point program, where they get a discount $d$ if their total traveling distance is beyond a constant $m$. This cost model is represented as $c_{\text{non-Markov}}(x_{<t}, x_t) = c_{\text{Markov}}(x_{t-1}, x_t) - d\mathbb{I}[\sum_{i=1}^{t-1} c_{\text{Markov}}(x_i, x_{i+1}) > m]$. Generally, under a budget constraint, dynamic costs require efficient nonmyopic planning; otherwise, the next decision may incur a high cost or fail to move beyond local optima.

## 3 METHOD

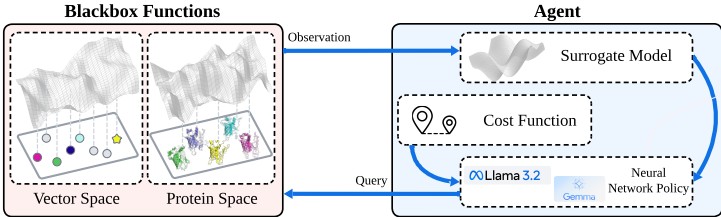

Figure 3: Illustration of the Optimization Process. The agent employs a surrogate model $f_\theta$ to approximate the complex black-box function $f$ in vector or protein spaces. A neural network policy $\xi : (x_{1:t}, y_{1:t}) \mapsto x_{t+1}$ generates the next queries to explore and exploit the optimization space. The next queries are generated by optimizing $\text{EHIG}(x_{1:L})$ objective which includes loss function $\ell(\cdot)$ computed from the action $a$ and surrogate model $f_\theta$, as well as cost function $c(\cdot)$.

### 3.1 BAYESIAN OPTIMIZATION IN DYNAMIC COST SETTINGS

DM employs an acquisition function to choose the next query. We study a general class of acquisition function based on decision-theoretic entropy, known as $H$-Entropy Search (HES) (DeGroot, 1962; Neiswanger et al., 2022) since many common acquisition functions, such as Knowledge Gradient (Frazier et al., 2009) and Expected Improvement (Šaltenis, 1971), can be considered a specific case of HES. This section briefly describes the multi-step lookahead variant of HES, showing how it can accommodate various dynamic cost structures $c$. We then describe the dynamic cost structure in detail. Lastly, we describe our procedure to optimize the variational version of the acquisition function by training a neural network policy.

For a prior $p(f)$ and a dataset $D_t = D_0 \cup \{(x_i, y_i)\}_{i=1}^t$, the posterior $\mathbb{H}_{\ell,c,\mathcal{A}}$-entropy and the expected $\mathbb{H}_{\ell,c,\mathcal{A}}$-information gain (EHIG) at step $t$ with loss function $\ell$, cost function $c$, action set $\mathcal{A}$, and Lagrange multiplier $\lambda$, and lookahead horizon $L$ is

$$\mathbb{H}_{\ell,c,\mathcal{A}}[f|D_t] = \inf_{a \in \mathcal{A}} \{\mathbb{E}_{p_t(f)}[\ell(f, a)] + \lambda c(x_{1:t}, a)\}$$

$$\text{EHIG}_t(x_{1:L}) = \mathbb{H}_{\ell,c,\mathcal{A}}[f|D_t] - \mathbb{E}_{p_t(y_{1:L}|x_{1:L})}\left[\mathbb{H}_{\ell,c,\mathcal{A}}[f|D_{t+L}]\right].$$

Following the $\mathbb{H}$-information gain heuristics, the decision-maker selects the query $x_{t+1} \in \mathcal{X}$ at each step $t$ to maximize the expected $\mathbb{H}$-information gain:

$$x_{1:L}^* = \underset{x_{1:L} \in \mathcal{X}^L}{\arg\sup} \, \text{EHIG}_t(x_{1:L}) = \underset{x_{1:L} \in \mathcal{X}^L}{\arg\sup} \left[-\mathbb{E}_{p_t(y_{1:L}|x_{1:L})}[\mathbb{H}_{\ell,c,\mathcal{A}}[f|D_{t+L}]]\right] \quad (1)$$

Under dynamic cost settings, *candidates in the query space $\mathcal{X}$ are not equally considered by the DM*. For instance, in the spotlight cost, the DM ignores candidates outside the spotlight radius to respect the cost structure. Lookahead allows the DM to consider candidates accessible in future steps based on their next query. We define the decision receptive field as the subset of the input space the DM considers. Lookahead expands this receptive field (see Figure 1) and enables the DM to "invest": choosing a suboptimal query now to access better outcomes (e.g., global optima) later. A longer lookahead horizon improves planning but exponentially increases decision variables and uncertainty, complicating the process. In the next section, we address this scalability challenge using a neural network policy.

## 3.2 NEURAL NETWORK POLICY OPTIMIZATION

As the lookahead horizon $L$ increases, optimization and integration dimensions grow, complicating the problem. We address this using variational optimization and pathwise sampling (see Appendix C.2). In nonmyopic decision-making, the number of decision variables scales with Monte Carlo samples, which depend on the number of paths $p$ and horizon length $T$. In the best case, where samples grow linearly with $T$, policy complexity is $\mathcal{O}(T)$. In the worst case, samples grow exponentially, increasing complexity to $\mathcal{O}(k^T)$, where $k$ is samples per step. To mitigate this, we employ a variational network, reducing the growth rate of decision parameters from exponential to constant with respect to the lookahead horizon. Variational optimization has been well studied and applied in various contexts, such as policy gradient methods (Schulman et al., 2017), VAEs (Kingma & Welling, 2014), and variational design of experiments (Foster et al., 2019). However, to the best of our knowledge, this approach has not yet been applied in the nonmyopic BO setting to reduce optimization complexity with respect to the lookahead horizon. From equation 1, we observe that for each lookahead step $l \in [L]$, the decision variable, $x_{t+l+1}^*$, is determined by the previous decision variables and corresponding observations, $(x_{1:t+l}, y_{1:t+l})$. This dependency can be modeled using a recurrent neural network (RNN) parameterized by $\xi \in \Xi$, which takes the history as input to predict the optimal next query: $\xi : (x_{1:t}, y_{1:t}) \mapsto x_{t+1}$. The corresponding posterior predictive $y_{t+1}$ can then be computed by $y_{t+1} \sim p_t(x_{t+1})$. Using pathwise sampling, this computation becomes $y_{t+1} = f(x_{t+1}^*)$, where $f \sim p_t(f)$. Thus, we can maintain gradients $\frac{\partial \text{EHIG}_t(x_{1:L})}{\partial \xi}$ for optimizing $x_{t+1} = \xi^*(x_{1:t}, y_{1:t})$ across the lookahead steps by applying the chain rule:

$$\frac{\partial \text{EHIG}_t(x_{1:L})}{\partial x_{t+L}} \frac{\partial x_{t+L}}{\partial \xi} + \frac{\partial \text{EHIG}_t(x_{1:L})}{\partial y_{t+L}} \frac{\partial y_{t+L}}{\partial x_{t+L}} \frac{\partial x_{t+L}}{\partial \xi}.$$

We can rewrite equation equation 1 as:

$$\xi^* = \underset{\xi \in \Xi}{\arg\inf} \left[ \mathbb{E}_{p_t(y_{1:L}|x_{1:L}, \xi)} \left[ \inf_{a \in \mathcal{A}} \left\{ \mathbb{E}_{p_{t+L}(f)}[\ell(f, a)] + \lambda c(x_{1:t}, x_{1:L}, a) \right\} \right] \right].$$

In our experiments, the variational network is trained using fantasized data points. Specifically, when optimizing step $t + 1$, we use the previously observed data points $(x_{1:t}, y_{1:t})$ to generate imagined lookahead data points $(x_{t+1:t+L}, y_{t+1:t+L})$ through an autoregressive process: $x_{t+l+1} = \xi_t(x_{1:t+l}, y_{1:t+l})$ and $y_{t+l+1} \sim p_{t+l}(x_{1:t+l})$. These fantasized data points are then used to compute the optimization objective and find the optimal $\xi_{t+1}$. Predicting the next query is an autoregressive process, making language models well-suited to be a variational network. Language models encompass various architectures, from classical LSTMs Hochreiter & Schmidhuber (1997) to modern transformer-based models like BERT Devlin et al. (2019) and decoder-only LLMs (e.g., GPT-4, LLaMA 3). As autoregressive models, they can also handle continuous time-series data Luo & Wang (2024). Here, previous queries serve as input, either directly or following a linear transformation. With $\mathcal{X}$ and $\mathcal{A}$ as continuous sets, language models can naturally function as policies.

In practical applications, query and action can be discrete, such as in drug design, where molecules are represented as strings (discrete characters). The final action may involve accepting or rejecting a generated sequence for wet lab testing. LookaHES extends to discrete cases by using a neural encoder to map tokens into continuous embeddings, similar to language modeling. Since the model outputs discrete tokens, differentiation in computing EHIG is challenging. This can be addressed using the reparameterization trick (Kingma & Welling, 2014) for small token sets or the REINFORCE algorithm (Williams, 1992), which leverages the log-derivative trick (Mohamed et al., 2020) to estimate gradients $\frac{\partial \text{EHIG}_t(x_{1:L})}{\partial \xi}$ efficiently. We provide the pseudo algorithm for our method in Algorithm 1.

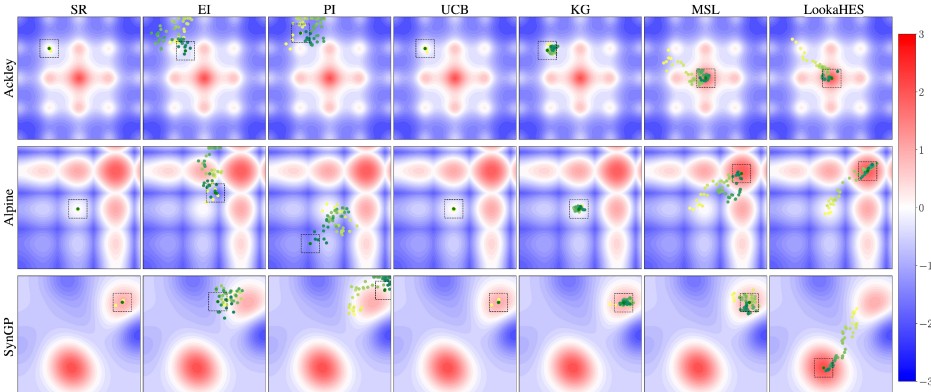

Figure 4: Queries across BO iterations with $\sigma = 0.0$ and $r$-spotlight cost. Yellow and green points indicate the initial position and final action, respectively. LookaHES reaches the global optimum, whereas the others tend to be trapped in local optima.

# 4 EXPERIMENTS

This section evaluates the performance and robustness of LookaHES. Our experiments aim to: (i) assess its efficiency in environments with dynamic query costs, (ii) benchmark it against established baselines on synthetic and real-world datasets, and (iii) demonstrate its advantages in performance and computational efficiency. Through these experiments, we address key research questions.

- **RQ1:** How does the proposed method compare to state-of-the-art myopic and nonmyopic acquisition functions in the continuous input domain under dynamic cost constraints?
- **RQ2:** Is it possible to apply the proposed method to problems with discrete input spaces?
- **RQ3:** How do aleatoric and epistemic noises, the quality of the surrogate model, and the lookahead horizon impact the performance of the proposed method?
- **RQ4:** Does 'optimism' in myopic methods lead to better performance than nonmyopic methods without optimism? Can this optimism be broadly applied to real-world problems?

We compare LookaHES with the six baselines implemented in BoTorch (Balandat et al., 2020) including Simple Regret (SR) (Zhao et al., 2023), Expected Improvement (EI) (Mockus, 1989), Probability of Improvement (PI) (Kushner, 1964), Upper Confidence Bound (UCB) (Srinivas et al., 2010), Knowledge Gradient (KG) (Frazier et al., 2009), and Multistep Tree (MSL) (Jiang et al., 2020b). Details of these baselines are presented in Appendix D. All acquisition function values are estimated via the quasi-Monte Carlo method with the Sobol sequence Balandat et al. (2020). We experiment with 4 cost functions: Euclidean, Manhattan, $r$-spotlight, and non-Markovian cost based on Euclidean distance. We use Sample Average Approximation with a base sample as a variance reduction technique that significantly improves the stability of optimization. To enhance the likelihood of convergence, we perform all optimizations using 64 restarts. The lookahead horizon is set to 20 for LookaHES and Multistep Tree. Each experiment is repeated with three random seeds. All experiments are conducted on an A100 GPU and 80GB memory.

## 4.1 CONTINUOUS OPTIMIZATION OF SYNTHETIC FUNCTIONS

To answer **RQ1**, we evaluate LookaHES on nine synthetic functions for global optimization in continuous vector spaces. The 2-dimensional functions, with their initial data points and maximum BO steps, include Ackley (50 samples, 100 steps), Alpine (100 samples, 50 steps), HolderTable (100 samples, 50 steps), Levy (100 samples, 50 steps), Styblinski-Tang (50 samples, 50 steps), and SynGP (25 samples, 50 steps). The SynGP function is generated from a 2D GP with a Radial Basis Function kernel, characterized by a length scale of $\sqrt{0.25}$ and a signal variance of 1. High-dimensional functions include Ackley4D (4D, 100 samples, 100 steps), Hartmann (6D, 500 samples, 100 steps), and Cosine8 (8D, 200 samples, 100 steps). Detailed descriptions of these functions

are available in (Bingham, Accessed 2024). We also apply LookaHES to a real-world problem in continuous space on identifying areas with the most light in NASA satellite images. The results of this experiment are presented in Appendix F. To further investigate the robustness of our approach, we conduct ablation studies on noise levels, initial data points, surrogate model kernels, lookahead horizon, and hyperparameter choices in the myopic acquisition function, as detailed in Appendix E.

The input of all functions is normalized in a hypercube $[0, 1]^d$, and the output is normalized to the range $-3$ to $3$. The global maximization of each function is at $3$, and the instantaneous regret of action $a$ is $3 - f^*(a)$. The outputs of all functions are observed with three levels of noise: $0\%$, $1\%$, and $5\%$. Gaussian Process is used as the surrogate model for these experiments. The variational neural network consists of a two-layer encoder, a Gated Recurrent Unit, and a three-layer decoder with exponential linear unit activations (64 hidden dimensions), optimized using Adam with a $10^{-3}$ learning rate. During inference, small noise from the von Mises-Fisher distribution (Fisher, 1953) is added to predicted queries to enhance exploration and acquisition function optimization. Figure 4 shows that myopic algorithms often converge to local maxima due to their short-term focus, while LookaHES considers future outcomes, guiding it toward the global maximum. With our improvements, the MSL method achieves a 20-step lookahead, matching LookaHES in outcomes. However, without the variational network, MSL optimizes directly on decision variables, limiting its real-world applicability. Figure 5 compares baseline methods and our approach in terms of final observed values under the highest noise level ($\sigma = 0.05$) across four cost functions.

***Summary:*** LookaHES consistently outperforms the myopic and is comparable to nonmyopic baselines on synthetic functions, highlighting the advantage of lookahead capabilities.

## 4.2    DISCRETE OPTIMIZATION OF PROTEIN FLUORESCENCE

We demonstrate the application of LookaHES to optimizing protein sequences (Elnaggar et al., 2023). The decision-making process involves determining whether to edit a given protein sequence. This experiment addresses **RQ2** with real-world problems. Figure 6 (left) provides a visualization of the protein space. If editing is chosen, the next step is to determine the position to be edited and select the new amino acid. We conduct a sequence of $T = 12$ edits to maximize the fluorescence level obtained from a wet lab experiment, given by the black-box oracle $f^* : \mathcal{X} \to \mathbb{R}$, which is expensive to query. We assume that $f^*$ has a parametric functional form on the feature space $\phi(x)$: $y = f_{\theta^*}(x) = g(x) + \alpha(\phi(x)^\top \theta^* + \epsilon)$, where $\theta^* \sim p(\theta) = \mathcal{N}(\mu, \Sigma)$, $\epsilon \sim \mathcal{N}(0, \sigma)$, $g(\cdot)$ is a synthetic function, and $\alpha$ is a scaling hyperparameter. In other words, we select a Gaussian prior for the model parameters.

We use the ProteinEA Fluorescence dataset (ProteinEA, 2024), which contains 21,445 training samples, to build the black-box oracle. We experiment with various featurization functions $\phi(\cdot)$, including Llama2 7B (Touvron et al., 2023), Llama3 8B (Meta, 2024), Mistral 7B (Jiang et al., 2023), Gemma 7B (GemmaTeam, 2024), ESM-2 650M, ESM-2 3B (Lin et al., 2022), and Llama-Molist-Protein 7B (Fang et al., 2024). Figure 17 shows validation results of the parametric black-box oracle with varying training sample sizes. Gemma 7B achieves the highest validation $R^2$ for predicting fluorescence, so we use it as the feature function.

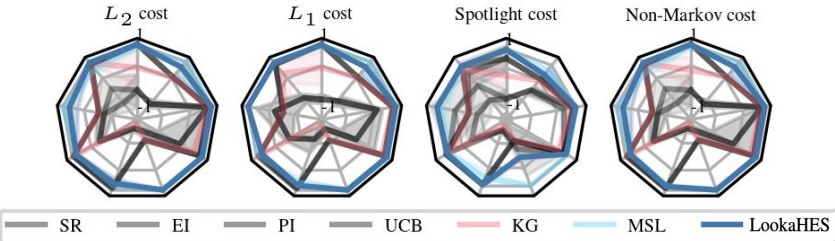

Figure 5: Final observed value at $\sigma = 0.05$. From noon (i.e., the north of each circle), counter-clockwise: Ackley, Ackley4D, Alpine, Cosine8, Hartmann, HolderTable, Levy, StyblinskiTang, SynGP. LookaHES consistently found global optimum across various cost structures.

Next, we construct our semi-synthetic protein space using a sequence from the ProteinEA Fluorescence. Specifically, we select a single sequence from the validation set consisting of 237 amino acids across 20 types. In this experiment, the protein designer can edit only one amino acid at a time across a maximum of 12 fixed positions and is limited to 2 possible amino acid types for each position. Under this setting, the protein space $\mathcal{X}$ contains $|\mathcal{X}| = 4096$ possible proteins. We then compute the fluorescence values for these proteins using the previously constructed oracle. The goal is to edit a starting protein to achieve the highest fluorescence, defined as $x_{max} = \arg\max_{x_i \in \mathcal{X}} \mathbb{E}_{\theta^* \sim p(\theta)} f_{\theta^*}(x_i)$. The starting protein, $x_0$, is chosen as the one with an edit distance of 12 from the protein with maximal fluorescence. Because each edit position can only accommodate two different tokens in this setup, there is only one possible starting protein. We choose $\alpha = 0.2$ and $g(x) = -0.005(d - 0.5)(d - 5)(d - 8)(d - 13.4)$, where $d = d_{edit}(x, x_0)$ represents the edit distance between the starting protein and a given protein $x$. We conduct an ablation study using a different starting protein and synthetic function $g(x)$, with results detailed in Appendix H.

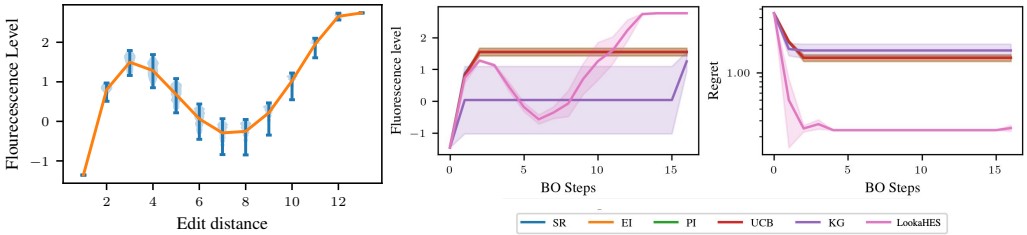

Figure 6: Fluorescence distribution by edit distance (left), observed fluorescence across BO steps (middle), and regret across BO steps (right). Myopic methods are trapped in local minima ( 1.5 fluorescence), while our 12-step nonmyopic approach anticipates the global maximum, achieving 2.7 fluorescence.

We use Bayesian linear regression as a surrogate to guide optimization. At each step $t \in [T]$, hyperparameters are estimated via maximum marginal log-likelihood, and the next mutation is selected by optimizing the variational network (Section 3). The mutation and fluorescence value are given by $x_{t+1} \sim p_{\xi_t}(x_t)$ and $f_{\theta^*}(x_{t+1})$, respectively. The optimization is batched, using Llama-3.2 3B as the variational network, with online and lookahead steps set to 16 and 12, respectively. Experiments are repeated with three seeds. The model is first supervised finetuned on random mutation data. During online iteration, we optimize the network with up to 768 gradient steps and 64 restarts for mutation selection. To improve efficiency, we modify Proximal Policy Optimization (PPO) by decoupling training and inference, using vLLM (Kwon et al., 2023) for lookahead rollouts. Network weights are transferred to vLLM after each gradient update. For constrained cost handling during lookahead sequence generation, we attempt regeneration up to 32 times. If regeneration is unsuccessful, we randomly mutate the most recent sequence with a 50% chance of retaining it. Figure 6 (middle) shows observed fluorescence levels, and (right) displays regret ($3 - f_{\theta^*}(a)$) across iterations. Additional details are in Appendix G.

*Summary:* LookaHES works well on discrete domains in protein editing, with superior performance to other myopic approaches, proving its effectiveness across diverse applications.

# 5 DISCUSSION, LIMITATIONS, AND FUTURE WORK

We propose LookaHES for the nonmyopic BO in dynamic cost settings. LookaHES incorporates dynamic costs and downstream utility, leading to more informed decision-making under uncertainty. By utilizing a neural network policy, it achieves scalability in planning multiple steps ahead. Experimental results demonstrate the superior performance of LookaHES compared to baseline methods on various benchmarks. However, the method has limitations. It requires a well-defined cost model upfront, which may not always be practical, and its performance relies heavily on a well-specified surrogate model. Future work should explore the impact of model misspecification on plan quality. Further advancements in adaptive modeling and cost estimation could enhance its robustness, expanding its applicability to even more complex decision-making scenarios.

ACKNOWLEDGEMENTS

NH acknowledges support by NSF 2302791 and 22298673 and Stanford HAI. SK acknowledges support by NSF 2046795 and 2205329, the MacArthur Foundation, Stanford HAI, and Google Inc.

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

## A  NOTATION

Table 2 is a glossary of the mathematical notation used in the paper.

Table 2: Glossary of mathematical notation

| Symbol | Description |
| --- | --- |
| $f^*$ | Black-box function |
| $p(f)$ | Prior distribution over the black-box function |
| $\theta$ | Random variable representing the parameters of the black-box function in the parametric form |
| $\theta^*$ | Optimal parameters of the black-box function in the parametric form |
| $\mathcal{X}$ | Input domain |
| $\mathcal{Y}$ | Output domain |
| $\xi$ | Parameters of the variational network |
| $D_t$ | $D_t = \{(x_i, y_i)\}_{i=1}^t$ Dataset acquired |
| $p_t(\cdot)$ | The posterior distribution conditioned on the data up to and including timestep $t$ |
| $c(\cdot, ..., \cdot)$ | $c : \mathcal{X}^k \to \mathbb{R}$, cost function depending on $k$-step history of query |
| $[T]$ | $\{1, ..., T\}$ |
| $\mathbb{H}_{\ell, \mathcal{A}}$ | The decision-theoretic entropy (DeGroot, 1962) corresponding to a loss function $\ell$ and an action set $\mathcal{A}$ |
| $L$ | Lookahead horizon |
| $T$ | Number of interactions with the environment |

## B  RELATED WORKS

**Nonmyopic Bayesian Optimization in the Dynamic Cost Setting**  Nonmyopic BO has been extensively explored in prior works (Osborne, 2010; González et al., 2016; Wu & Frazier, 2019; Jiang

et al., 2020a; Lee et al., 2020; 2021; Astudillo et al., 2021; Folch et al., 2022; Belakaria et al., 2023; Jiang et al., 2020b). These studies focus on converting a nested, multi-step planning problem into a single, high-dimensional optimization problem that can be solved efficiently with quasi-Monte Carlo sampling and gradient-based optimization. The advantage of having a lookahead mechanism is enlarging the receptive field—the area the decision maker can see prior to making a decision (Figure 1). This approach has gained traction in cost-aware and budget-constrained BO (Astudillo et al., 2021; Lee et al., 2021), where nonmyopic planning is crucial. However, a notable challenge with this methodology is its limited scalability when extending the lookahead horizon, primarily due to the exponential increase in the number of decision variables. Our approach introduces a novel combination of Thompson sampling (Thompson, 1933) with the extensive generalization capabilities of a variational network, significantly enhancing the computational efficiency of nonmyopic BO. Utilizing neural networks for variational inference and optimization is not a new concept (Kingma & Welling, 2014; Amos, 2023). For instance, Deep Adaptive Design (Foster et al., 2021; Ivanova et al., 2021) is a parallel line of research from the Bayesian optimal experimental design literature, which has a related but distinct objective to reduce the uncertainty of model parameters as opposed to the global optimization objective in BO. The authors concentrate on reducing the computational demands during deployment and determining the most informative experimental designs by upfront offline optimization of a neural network to amortize the design cost. Our approach diverges by aligning more closely with the principles of online Bayesian optimization. Here, the primary objective extends beyond mere information acquisition to encompass the pursuit of global optimization. Our approach, which employs adaptive decision-making through online policy optimization, could be more robust than offline methods, particularly when the approximation of the reward function changes significantly between queries. This robustness arises because the online policy is updated at each BO step while the offline methods rely on transferring knowledge from learned offline data (Nguyen-Tang & Arora, 2024). The distinctive feature of our study is implementing a pathwise, Thompson sampling-based nonmyopic acquisition function, which significantly reduces the computational cost of the iterative posterior sampling approach in (Jiang et al., 2020b). Additionally, we present detailed comparisons of related works on different cost structures in Appendix 2.2.

**Variational Policy Optimization in Complex Action Spaces**  In many real-world applications, decision-makers must take actions that are complex and subject to semantic constraints. Semantic constraints refer to rules or relationships that restrict the set of valid actions based on their meanings, dependencies, or contextual appropriateness. For example, in biological sequence design, semantic constraints may ensure that the mutated sequence is valid and does not unfold the protein (i.e., protein denaturalization). Recent RL research has addressed environments with such actions, which are challenging due to two main reasons: (i) the large number of potential actions (Hubert et al., 2021; Zhang et al., 2024), and (ii) the complex semantics (Carta et al., 2023) underlying each action, making them difficult to capture. Recent studies have shown that modern LLMs can effectively model semantic actions and be fine-tuned with feedback from the environment (Zhu et al., 2024; Zhang et al., 2024; Zhuang et al., 2024; Hazra et al., 2024). Several papers demonstrate that using LLMs as policy models in reinforcement learning leads to better outcomes (Palo et al., 2023; Zhuang et al., 2024; Hazra et al., 2024). In the field of NLP, a chatbot such as ChatGPT can be viewed as a decision-making process where the underlying LLM must understand user questions or requests to provide appropriate responses. These actions are complex and semantically rich, as even a single word can alter the meaning of a sentence. Consequently, RL methods like proximal policy optimization (Schulman et al., 2017) have been applied to refine the abilities of language models.

**Multi-turn Training Framework for LLMs**  Multi-turn conversations have been shown to be more effective for managing entire dialogues (Zhou et al., 2024). This approach can be viewed as a nonmyopic RL method that trains LLMs to achieve better conversational outcomes. Unfortunately, current RL training frameworks for LLMs, such as TRL (von Werra et al., 2020), OpenRLHF (Hu et al., 2024), LlamaFactory (Zheng et al., 2024), and Nemo (Harper et al., 2019), primarily focus on single-turn conversations. As a result, they are not suited for multi-turn conversation training. When using these frameworks, multi-turn conversations must be divided into individual single turns, which limits the LLM's ability to manage the overall outcome of a conversation effectively.

## C   METHOD DETAIL

### C.1   PSEUDO-ALGORITHM

---

**Algorithm 1:** LookaHES Algorithm

---

**Input**  : Black-box function $f^*$
   Initial dataset $D_0$
   Loss function $\ell(f, a)$
   Cost function $c(x_{t-1}, x_t)$
   Lagrange multiplier $\lambda$
   Maximum number of query $T$
   Number of lookahead step $L$
**Output:**  Final dataset $D_T$
**for** $t = 1 \rightarrow T$ **do**
$\quad f_\theta \sim p(\theta|D_{t-1})$
$\quad \xi^* = \arg\inf_{\xi \in \Xi} \left[ \mathbb{E}_{p_t(y_{1:L}|x_{1:L},\xi)} \left[ \inf_{a \in \mathcal{A}} \left\{ \mathbb{E}_{p_{t+L}(f)}[\ell(f_\theta, a)] + \lambda c(x_{1:t}, x_{1:L}, a) \right\} \right] \right]$
$\quad x_t \leftarrow \xi^*(x_{1:t-1}, y_{1:t-1})$
$\quad y_t \leftarrow f^*(x_t)$
$\quad D_t = D_{t-1} \cup \{(x_t, y_t)\}$
**end**

---

### C.2   PATHWISE SAMPLING

When the surrogate model is a Gaussian Process (GP), the Monte Carlo method is employed to evaluate the posterior predictive distribution. In prior works, this is done via iterative sampling of the following factorized distribution: $p(y_{1:T}|x_{1:T}, D_0) = \prod_{t=1}^{T} p(y_t|x_t, x_{<t}, y_{<t}, D_0)$. The posterior predictive distribution at the $t$-th step, denoted as $p(y_t|x_t, x_{<t}, y_{<t}, D_0)$, can be approximated by generating $k$ samples of $y_t$ from the GP model. In general, the value of $k$ varies depending on the specific problem. At iteration $t$, suppose that we always sample $k$ samples from the posterior predictive distribution. The number of $y_t$ is $k^t$. This number quickly explodes exponentially with the length of the lookahead horizon (Figure 7). The GP posterior predictive sampling process involves computing the square root of the covariance matrix, which is typically done via Cholesky decomposition. The complexity of this process is proved as $\mathcal{O}(n^3)$ for exact GP or $\mathcal{O}(m^3)$ for approximate GP where $n$ is the total number of samples in the training dataset and $m < n$ is the number of inducing samples (Quiñonero-Candela & Rasmussen, 2005; Wilson et al., 2020). This evidence shows that the complexity for sampling posterior predictive distribution at step $t$-th is at least $\mathcal{O}(k^t m^3)$. One variant of this procedure that can reduce the complexity is limiting the number of sampling samples for posterior predictive approximation at further lookahead steps. For instance, at each step $t > 1$, we can set $k_{t>1} = \max(k_1/2^t, 1)$, where $k_1$ is the predefined number of samples at the first lookahead step. In these cases, we can observe that $\exists \tau : \forall t > \tau, \prod_{t=1}^{T} k_t = K$, where $K$ is a constant. Subsequently, the complexity at step $t$-th can be reduced to $\mathcal{O}(\prod_{t=1}^{T} k_t m^3) = \mathcal{O}(K m^3) = \mathcal{O}(m^3)$.

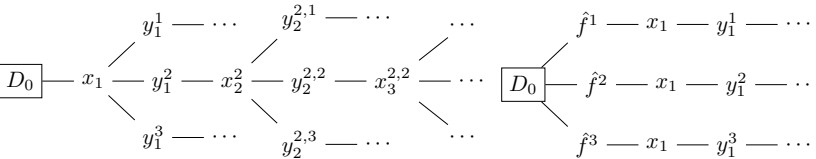

Figure 7: Posterior predictive sampling (left) and Pathwise sampling (right)

To mitigate the high complexity of above sampling process, we employ the following factorization: $p(y_{1:T}|x_{1:T}, D_0) = \int p(y_{1:T}|x_{1:T}, f)p(f|D_0)\,\mathrm{d}f = \int p(f|D_0)\prod_{t=1}^{T} p(y_t|x_t, f)\,\mathrm{d}f$. The function $f$ is drawn from the prior distribution and path-wise updated via Matheron's rule. For the $h$ path, consisting of $T$ steps each, the sampling can be done with complexity $\mathcal{O}(h \times T)$. We can approximate

the integral arbitrarily well with higher $h$. The gain comes from the fact that we do not need to iteratively compute $K_{m,m}^{-1}$ as in fantasization. If we did, the complexity, with the same number of samples, would be $\mathcal{O}(h \times (T-1)^3)$. This can be done in linear complexity w.r.t. to the number of samples. The complexity of sampling a posterior $\hat{f}$ from $p(f|D_0)$ can be considered as $\mathcal{O}(C)$, where $C$ is a constant because the number of samples in $D_0$ is unchanged. Then, computing $y_t$ for approximate posterior predictive $p(y_t|x_t, \hat{f})$ can be done by $y_t = \hat{f}(x_t)$, which has complexity of $\mathcal{O}(1)$. Using the same technique as limiting the number of sampling samples, the complexity approximating posterior predictive at any lookahead step is $\mathcal{O}(K)$. Thus, the total complexity at each step $t$-th is $\mathcal{O}(C + K)$. Figure 7 (right) visualizes the concept of this method.

## D    DETAILS OF BASELINES

- Simple Regret (SR) (Zhao et al., 2023) measures the regret or loss in performance between the updated model and the model that would have resulted if the optimal sample had been selected for annotation during the active learning process instead.

- Expected Improvement (EI) (Mockus, 1989) is used to evaluate the usefulness of candidate samples by estimating the expected gain in the performance of a model.

- Probability of Improvement (PI) (Kushner, 1964) calculates the probability of a candidate sample improving the performance of a model compared to the current best sample.

- Upper Confidence Bound (UCB) (Srinivas et al., 2010) balances exploration and exploitation by selecting candidate samples with high uncertainty and high potential for improvement based on the upper confidence bound of their predicted performance.

- Knowledge Gradient (KG) (Frazier et al., 2009) quantifies the expected improvement in the objective function value resulting from evaluating a specific point. It considers the uncertainty of the model predictions and the potential benefit of obtaining additional information about the objective function.

- Multistep Tree (MSL) (Jiang et al., 2020b), which can look up to four steps ahead, is constrained by computational costs. We reimplement this acquisition function using Pathwise sampling, enabling a lookahead horizon of up to 20 steps.

## E    ABLATION STUDIES ON SYNTHETIC FUNCTIONS

### E.1    VARYING THE OBSERVATION NOISE LEVEL

To answer the **RQ3** in terms of the impact of aleatoric noise on BO methods, we performed an ablation study by varying the observation noise levels at 0%, 1%, and 5%. A comprehensive comparison of LookaHES against baseline approaches was conducted across nine synthetic functions, incorporating all cost structures and noise levels. As shown in Figure 8, LookaHES consistently achieves superior performance across all cost structures and noise settings.

### E.2    VARYING THE NUMBER OF INITIAL SAMPLES

In many scenarios, limited data availability at the start of an optimization process leads to a poorly constructed surrogate model, resulting in high epistemic uncertainty. Understanding the behavior of BO methods under such conditions is crucial for planning appropriate actions. Specifically, we investigated how varying the number of initial samples affects the optimization process to answer the **RQ3**. The analysis was conducted across three environments: Ackley, Alpine, and SynGP, with evaluations performed at three different levels of initial samples (Figure 9). Our results indicate that with fewer initial points, the GP surrogate model struggles to accurately approximate the ground-truth function, thereby increasing the likelihood of suboptimal outcomes across both myopic and nonmyopic methods.

To address this issue, we enhance the diversity of the generated outputs and introduce a warm-up phase for the amortized network parameters during each BO iteration. Specifically, we set the $\kappa$ concentration hyperparameter of the von Misher-Fisher distribution (distribution of noises added to output) to 0, which promotes a more diverse range of outputs. For the warm-up phase, the amortized

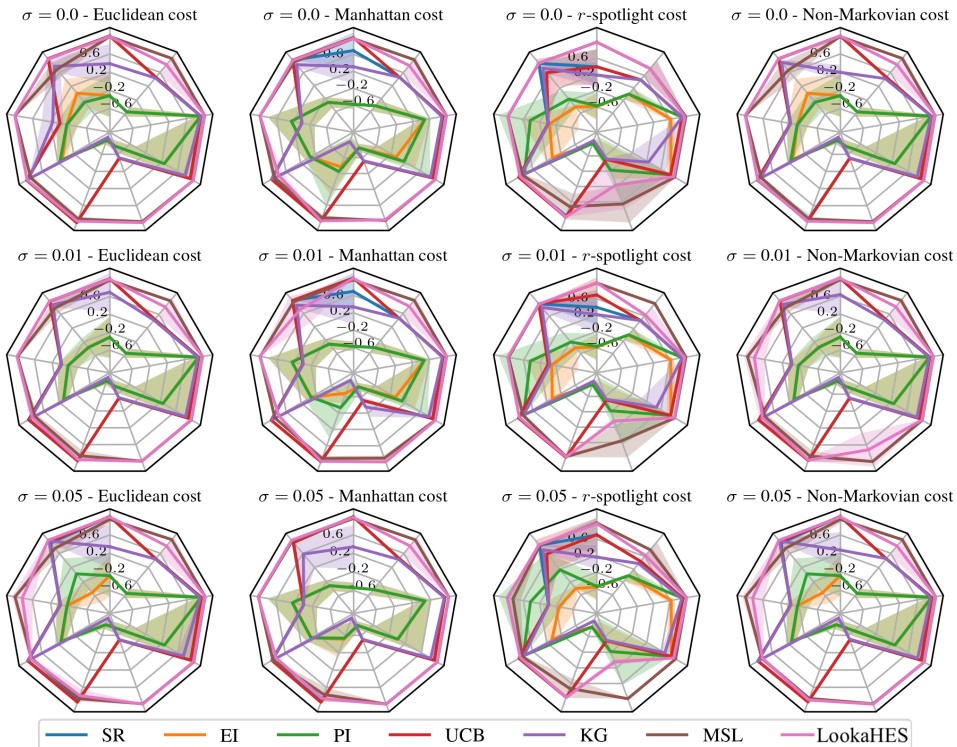

Figure 8: Final observed value. Starting from noon, counter-clockwise: Ackley, Ackley4D, Alpine, Cosine8, Hartmann, HolderTable, Levy, StyblinskiTang, SynGP. We observe that LookaHES consistently achieves the global optimum across various types of cost structures and noise levels

network is initialized with randomly generated data points, preventing the generated outputs from being confined to a local region. This approach ensures broader coverage of the receptive field, facilitating better exploration. The refined results are presented in Figure 10, demonstrating the effectiveness of this strategy within the SynGP environment.

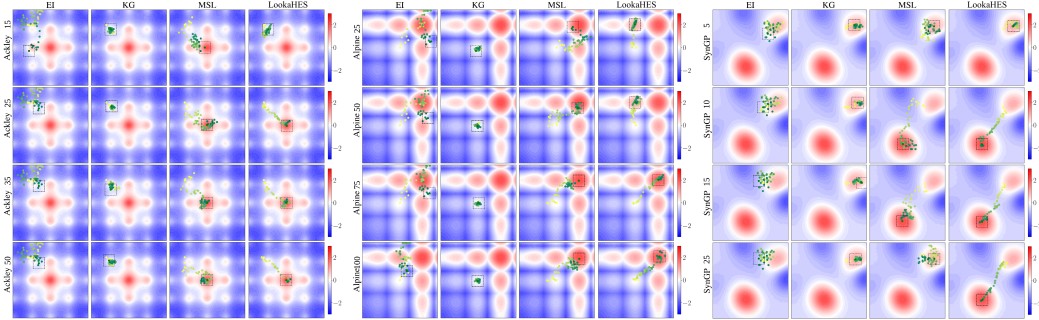

Figure 9: Comparison of performance between LookaHES and baselines with different numbers of initial samples. The yellow points indicate the starting positions, while the green points represent the final actions. From top to bottom, the Ackley function is evaluated with 15, 25, 35, and 50 initial samples; the Alpine function with 25, 50, 75, and 100 initial samples; and the SynGP function with 5, 10, and 15 initial samples. With a small number of initial samples, all methods tend to fail to find the global optimum due to poor surrogate models.

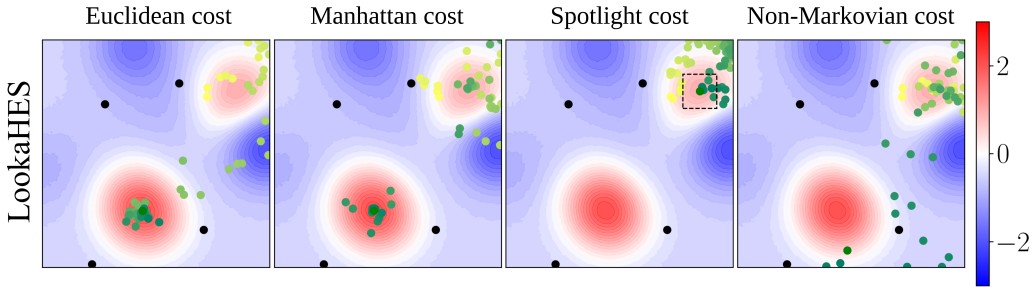

Figure 10: SynGP environment with 5 initial points and our diversity-enhanced BO method. The black points are data. The yellow points indicate the starting positions, while the green points represent the final actions.

### E.3 VARYING THE CHOICE OF GAUSSIAN PROCESS KERNEL

Since our surrogate models are GPs, their quality is influenced not only by the number of data points but also by the choice of kernel. In this section, we address the **RQ3**, focusing on the impact of kernel selection on GP performance. To evaluate this, we tested different kernel functions, including the Radial Basis Function (RBF) kernel and the Matérn kernel with $\nu = 1.5$, across three functions: Ackley, Alpine, and SynGP. Figure 11 visualizes the ablation results. This ablation demonstrates that with any well-fitted kernel, the nonmyopic approach can achieve the global optimum.

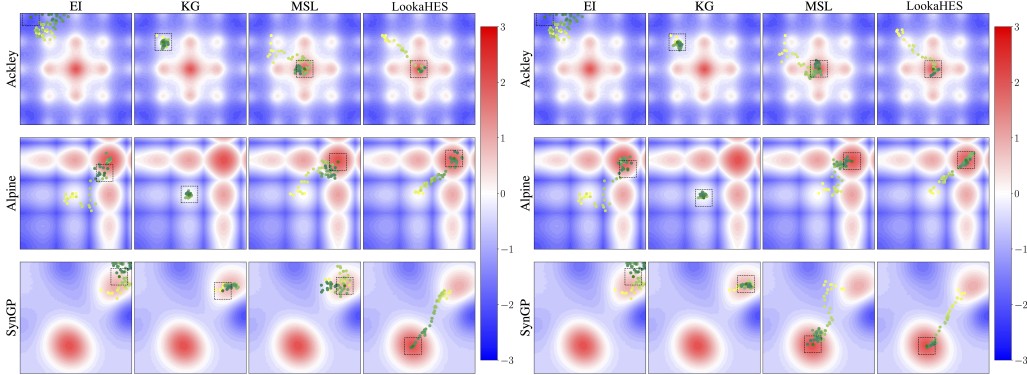

Figure 11: Comparison of performance between LookaHES and baselines with different kernels for the surrogate model (RBF on the left, and Matern on the right). The yellow points indicate the starting positions, while the green points represent the final actions. The performance of LookaHES is not affected by the choice of kernel for the surrogate model as long as the surrogate model can approximate the target function effectively.

In our synthetic experiments, we do not include an ablation study on the Bayesian linear regression model as it is unsuitable for accurately approximating the non-linear target functions. To demonstrate this limitation, we compared the posterior surface generated by Bayesian linear regression with those of other kernel-based methods, as shown in Figure 12. These results confirmed its inadequacy, leading us to exclude it from our ablation study.

### E.4 VARYING THE LENGTH OF THE LOOKAHEAD HORIZON

To answer **RQ3** on the benefit of a large lookahead horizon, we included experimental results on the ablation of the number of lookahead steps in Figure 13. These results illustrate the relationship between the number of lookahead steps and the robustness of the optimization, providing insights into how the performance of our approach varies with different horizon lengths. Specifically, with a smaller lookahead horizon, the probability of being trapped by local optima increases, leading to suboptimal optimization in all nonmyopic methods.

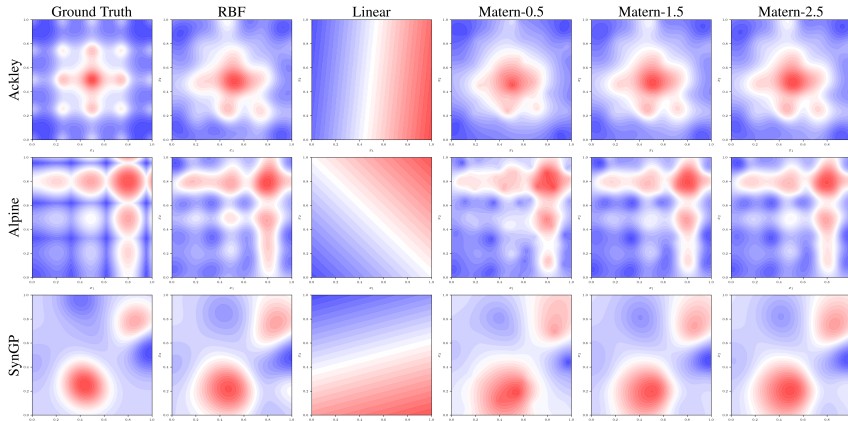

Figure 12: Comparison of posterior surfaces of different kernels on Ackley, Alpine, and SynGP function. Using Bayesian linear regression (the third column) resulted in a wrong approximation of the ground truth functions.

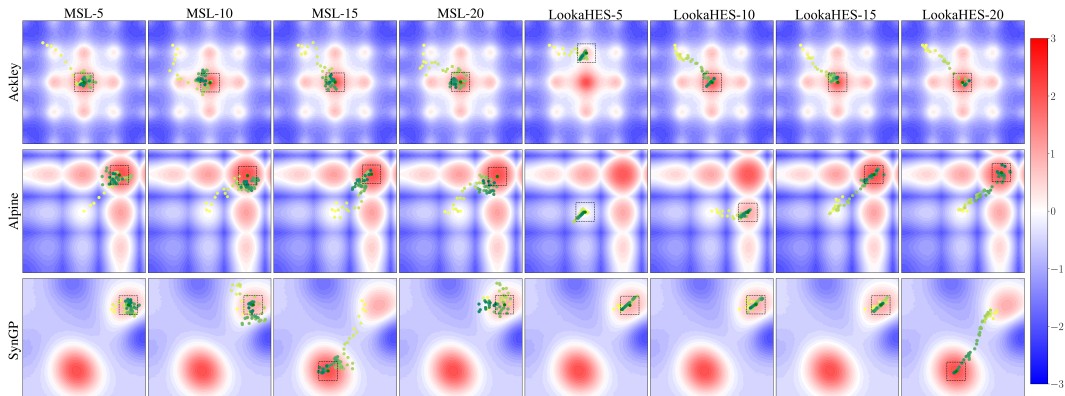

Figure 13: Comparison of LookaHES and nonmyopic baseline at 5, 10, 15, and 20 lookahead steps. The yellow points indicate the starting positions, while the green points represent the final actions. With fewer lookahead steps, nonmyopic methods tend to fail to find the global optimum, demonstrating the benefit of having a longer lookahead horizon.

The experiments conducted across various scenarios highlight the robustness and effectiveness of our proposed method in handling different challenges in Bayesian optimization. We observed that varying observation noise levels (0%, 1%, and 5%) had minimal impact on the performance of LookaHES compared to baseline approaches, with LookaHES consistently achieving the global optimum across all cost structures and noise settings. Additionally, we found that limited initial samples led to suboptimal performance due to high epistemic uncertainty, but introducing a diversity-enhancing strategy and a warm-up phase improved the results. Kernel selection also played a crucial role, and LookaHES demonstrated strong performance regardless of the kernel choice as long as the surrogate model could approximate the target function effectively. Finally, our results show that incorporating a larger lookahead horizon significantly improves the optimization process, reducing the likelihood of being trapped in local optima.

*Summary:* Our proposed method demonstrated robustness to aleatoric noise, maintaining strong performance even with a 5% noise level. High epistemic uncertainty from limited initial samples hindered performance, but strategies like diversity enhancement and warm-up phases mitigated this issue. Additionally, effective surrogate model design and a larger lookahead horizon were crucial, enhancing optimization by avoiding local optima and improving convergence.

## E.5 VARYING THE ACQUISITION FUNCTION HYPERPARAMETERS

Myopic acquisition functions, such as UCB, rely on "optimism" during optimization. This means they prioritize exploration with the expectation that querying enough points may eventually uncover the optimum. In this section, we investigate whether this "optimism" can outperform lookahead methods, addressing **RQ4**. In the case of the UCB acquisition function, the degree of optimism is controlled by the $\beta$ hyperparameter: smaller values of $\beta$ emphasize exploitation, while larger values encourage exploration. With sufficiently large $\beta$, the standard deviation term dominates the mean term, leading to decisions driven by the most uncertain areas. To further illustrate the impact of large $\beta$, we conducted additional experiments with $\beta$ values ranging from 0.1 to 1000 on nine synthetic functions. In Figure 14 we highlight the behavior of UCB when increasing $\beta$.

We also provide the value of the final action, normalized to range from -1 to 1, where -1 represents the worst outcome and 1 is the best in Table 3. These empirical results further illustrate that the large $\beta$ value can encourage the decision-maker to make queries that highly prioritize exploration. As illustrated in the above figure and table, such exploration are typically myopic and unplanned, and consequently, the decision maker typically misses the global optima or overexplore the un-promising region. We also want to note that in our experiment, no single $\beta$ outperformed others in all settings: for example, $\beta = 10$ works well for Ackley, but does not work for other functions. Indeed, choosing the value of $\beta$ for UCB before running the online experiment is nontrivial in practice.

Table 3: Comparison of final action value of LookaHES with 20-step lookahead and UCB with various $\beta$ value

| Method | Ackley | Ackley4D | Alpine | Cosine8 | Hartmann | HolderTable | Levy | StyblinskiTang | SynGP |
|---|---|---|---|---|---|---|---|---|---|
| LookaHES | $0.97 \pm 0.03$ | $0.97 \pm 0.02$ | $\mathbf{0.99 \pm 0.0}$ | $0.93 \pm 0.01$ | $0.96 \pm 0.03$ | $0.05 \pm 0.08$ | $\mathbf{0.95 \pm 0.0}$ | $\mathbf{1.0 \pm 0.0}$ | $\mathbf{0.63 \pm 0.25}$ |
| UCB ($\beta = 0.1$) | $0.7 \pm 0.4$ | $0.68 \pm 0.44$ | $-0.01 \pm 0.01$ | $0.96 \pm 0.01$ | $0.95 \pm 0.02$ | $-0.49 \pm 0.01$ | $0.87 \pm 0.0$ | $0.91 \pm 0.0$ | $0.45 \pm 0.01$ |
| UCB ($\beta = 0.5$) | $0.4 \pm 0.41$ | $0.68 \pm 0.44$ | $-0.01 \pm 0.01$ | $\mathbf{0.97 \pm 0.01}$ | $0.94 \pm 0.04$ | $-0.5 \pm 0.01$ | $0.87 \pm 0.0$ | $0.91 \pm 0.0$ | $0.45 \pm 0.01$ |
| UCB ($\beta = 1$) | $0.4 \pm 0.41$ | $0.67 \pm 0.43$ | $-0.01 \pm 0.01$ | $0.96 \pm 0.02$ | $0.95 \pm 0.03$ | $-0.49 \pm 0.01$ | $0.87 \pm 0.0$ | $0.91 \pm 0.0$ | $0.45 \pm 0.01$ |
| UCB ($\beta = 2$) | $0.4 \pm 0.41$ | $0.68 \pm 0.44$ | $-0.01 \pm 0.01$ | $0.97 \pm 0.02$ | $0.95 \pm 0.03$ | $-0.49 \pm 0.01$ | $0.87 \pm 0.0$ | $0.91 \pm 0.0$ | $0.45 \pm 0.01$ |
| UCB ($\beta = 5$) | $0.7 \pm 0.43$ | $0.98 \pm 0.01$ | $-0.01 \pm 0.01$ | $0.96 \pm 0.03$ | $\mathbf{0.97 \pm 0.02}$ | $-0.49 \pm 0.01$ | $0.87 \pm 0.0$ | $0.91 \pm 0.0$ | $0.45 \pm 0.01$ |
| UCB ($\beta = 10$) | $\mathbf{1.0 \pm 0.01}$ | $0.98 \pm 0.01$ | $-0.01 \pm 0.01$ | $0.97 \pm 0.02$ | $0.96 \pm 0.03$ | $-0.28 \pm 0.32$ | $0.87 \pm 0.0$ | $0.91 \pm 0.0$ | $0.45 \pm 0.01$ |
| UCB ($\beta = 20$) | $0.71 \pm 0.35$ | $0.87 \pm 0.17$ | $0.15 \pm 0.24$ | $0.97 \pm 0.04$ | $0.97 \pm 0.03$ | $-0.03 \pm 0.34$ | $0.87 \pm 0.0$ | $0.91 \pm 0.0$ | $0.45 \pm 0.01$ |
| UCB ($\beta = 50$) | $0.55 \pm 0.65$ | $\mathbf{0.99 \pm 0.01}$ | $0.73 \pm 0.36$ | $0.88 \pm 0.04$ | $0.94 \pm 0.03$ | $0.81 \pm 0.6$ | $0.87 \pm 0.0$ | $0.91 \pm 0.0$ | $0.45 \pm 0.01$ |
| UCB ($\beta = 100$) | $0.45 \pm 0.41$ | $0.47 \pm 0.33$ | $0.92 \pm 0.09$ | $0.71 \pm 0.06$ | $0.95 \pm 0.03$ | $0.9 \pm 0.81$ | $0.86 \pm 0.01$ | $0.94 \pm 0.04$ | $0.45 \pm 0.01$ |
| UCB ($\beta = 200$) | $-0.34 \pm 0.38$ | $-0.72 \pm 0.06$ | $0.15 \pm 0.46$ | $0.62 \pm 0.07$ | $0.81 \pm 0.12$ | $\mathbf{0.76 \pm 0.23}$ | $0.93 \pm 0.05$ | $0.93 \pm 0.1$ | $0.22 \pm 0.32$ |
| UCB ($\beta = 500$) | $-0.12 \pm 0.13$ | $-0.33 \pm 0.3$ | $-0.43 \pm 0.48$ | $0.13 \pm 0.18$ | $-0.02 \pm 0.61$ | $0.27 \pm 1.1$ | $0.82 \pm 0.12$ | $0.96 \pm 0.06$ | $0.61 \pm 0.53$ |
| UCB ($\beta = 1000$) | $0.06 \pm 0.5$ | $-0.6 \pm 0.24$ | $-0.2 \pm 0.66$ | $-0.18 \pm 0.1$ | $-0.52 \pm 0.43$ | $-0.41 \pm 0.32$ | $0.84 \pm 0.06$ | $0.92 \pm 0.07$ | $0.5 \pm 0.53$ |

Our experiments demonstrate that while large values of the $\beta$ hyperparameter in the UCB acquisition function prioritize exploration, they often lead to myopic and unplanned exploration, causing the decision-maker to miss the global optimum or over-explore unpromising regions. The performance of UCB varies across different functions, with no single $\beta$ value outperforming others universally, highlighting the challenge of selecting an optimal $\beta$ before running the experiment. This underscores the complexity of using optimism in myopic methods compared to more structured lookahead approaches.

> ***Summary:*** Optimism in myopic methods, such as using a large $\beta$ in the UCB acquisition function, can lead to unplanned exploration and suboptimal performance, as it risks missing the global optimum or over-exploring unpromising areas. While this optimism may occasionally benefit specific scenarios, its lack of consistency across functions limits its broad applicability to real-world problems compared to more structured, nonmyopic approaches.

## E.6 SYNTHETIC FUNCTIONS WITH DISCRETE INPUT

In many practical scenarios, data domains exhibit discrete attributes, such as those encountered in natural language processing or chemical molecular structures. To answer the **RQ2** and demonstrate the efficiency of LookaHES within discrete spaces, we conducted an additional experiment. In this experiment, we utilized the SynGP environment. However, in this case, we discretized the domain of each dimension into $C$ categories, such that the design variable belongs to the set $\mathbb{C}^d$, where $d$ represents the number of design dimensions. The categorical variables are represented in a one-hot encoding format. At time step $t$, the design variable $x_t$ is a matrix of size $C \times d$, with $x_t = [x_t^i]_{i=1}^d$, where $x_t^i \in \mathbb{C}$ denotes the one-hot encoded vector for the $i$-th dimension.

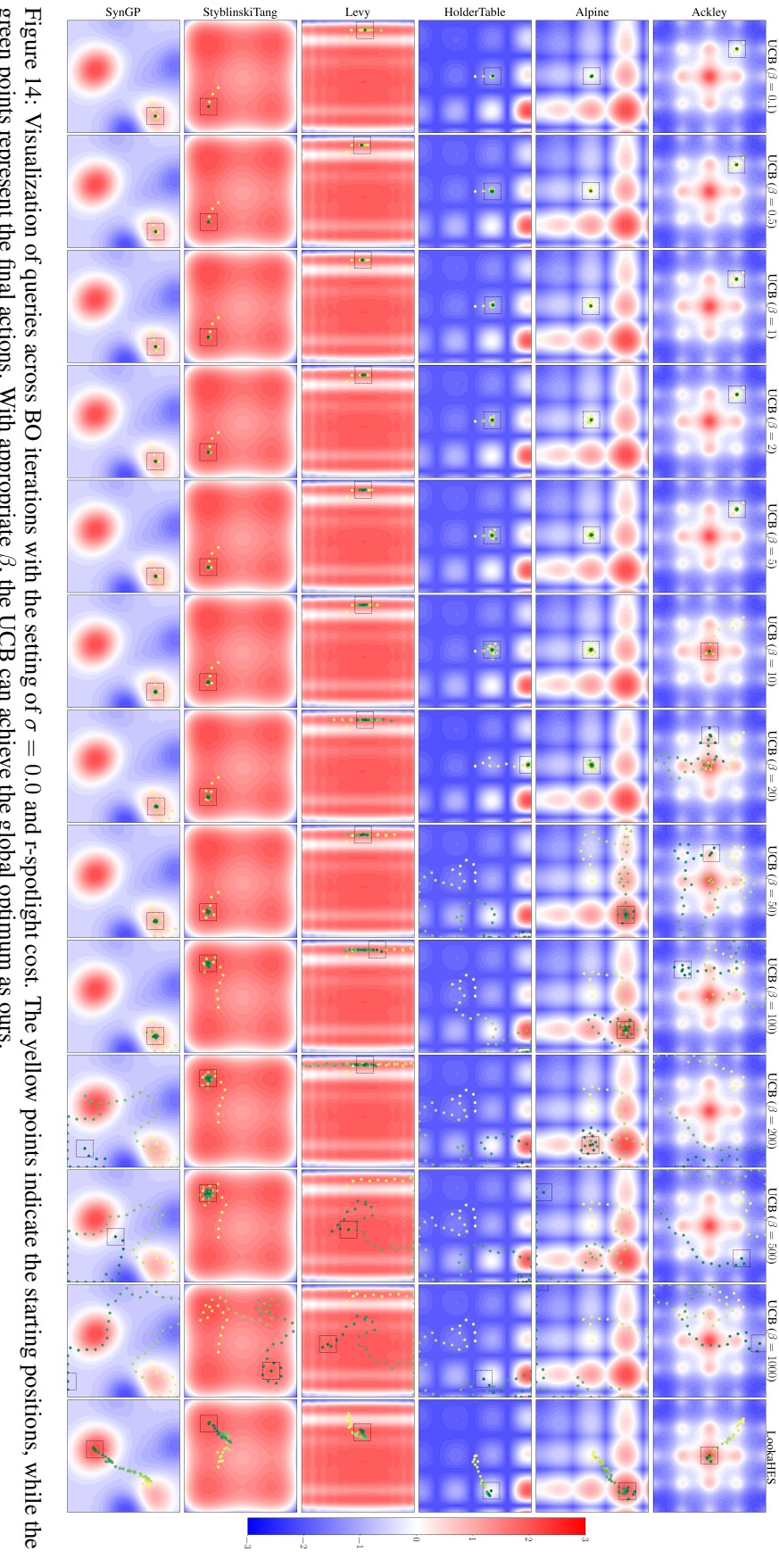

Figure 14: Visualization of queries across BO iterations with the setting of $\sigma = 0.0$ and r-spotlight cost. The yellow points indicate the starting positions, while the green points represent the final actions. With appropriate $\beta$, the UCB can achieve the global optimum as ours.

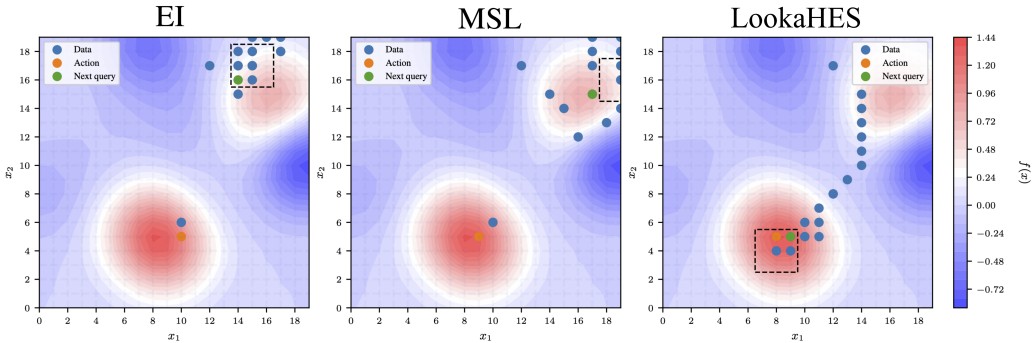

Figure 15: Results of discrete setting on SynGP. From left to right: EI, MSL, Our

To adapt our variational network to this discrete domain, we modify its architecture as follows. To compute the representation $x_t'$ of the design variable for the variational network, each dimension $x_t^i$ is passed through a linear transformation, followed by a sum-pooling operation across dimensions: $x_t' = \sum_{i=1}^{d} W x_t^i$. The output of the variational network is a probability vector corresponding to a categorical distribution. To ensure gradient propagation during optimization, we sample the most likely category using the Straight-through reparameterization technique (Bengio et al., 2013).

The MSL acquisition function in discrete domains does not benefit from gradient-based optimization methods to minimize the loss function. As a result, MSL is implemented using a multistep combinatorial search in this experiment. This approach significantly increases computational complexity, especially when dealing with a large number of categories. For instance, in the SynGP environment, where an input variable has 20 categories and the MSL algorithm is configured with $L = 4$ lookahead steps, the number of combinations to explore is given by $(C^d)^L = (20^2)^4 = 25.6 \times 10^9$, leading to an exponentially large search space. Given the constraints of limited computational resources, we resort to a random search with a budget of 2000 possible design variable configurations.

Figure 15 presents a comparative analysis of the performance of the myopic acquisition function EI, MSL, and our proposed acquisition function within the discrete SynGP after 18 BO iterations. As shown, LookaHES outperforms the other acquisition functions. This experiment thus highlights the robustness of our approach in addressing optimization problems in discrete domains.

## F  DETAILS OF NIGHT LIGHT EXPERIMENTS

To demonstrate the applicability of LookaHES in real-world continuous environments, we conducted experiments on human travel optimization within a 2D continuous domain. This experiment addresses **RQ1** in the context of continuous domains. Specifically, we used a 2016 grayscale image of night lights in Georgia and South Carolina, sourced from NASA's Earth Observatory, with a resolution of $1000 \times 1000$ pixels. The data is online accessible at `https://earthobservatory.nasa.gov/features/NightLights`. To facilitate the optimization of the GP surrogate model and avoid numerical issues due to image noise, we applied a stack blur with a radius of 40 to the image. The pixel values, ranging from 0 to 255, were normalized to a range of $-3$ to 3. The image width and height were normalized to a range of 0 to 1. We apply LookaHES and baselines with spotlight cost ($r = 0.1$) and Euclidean cost. Figures 16 show the results of LookaHES and baselines on the spotlight and Euclidean cost, respectively. In this environment, nonmyopic methods demonstrated their advantage in lookahead capability. Notably, LookaHES showed its effectiveness in directly reaching the global optimum, rather than querying around sub-optimal locations before approaching the global optimum as MSL.

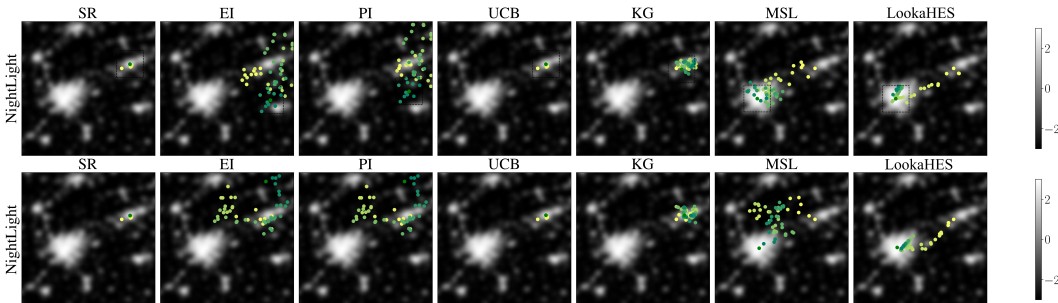

Figure 16: Visualization of different methods on NASA night light images in the case of spotlight cost (top row) and Euclidean cost (bottom row)

# G    DETAILS OF PROTEIN SEQUENCE DESIGN EXPERIMENTS

## G.1    ORACLE GOODNESS OF FIT

We assess the Oracle model's performance with increasing training data to gauge how well our semi-synthetic setting approximates real data. The Oracle is a Bayesian linear regression model that takes protein embeddings from various large language models, ranging from Llama-2 7B (a general-purpose model) to ESM-2 3B (a protein-specific model). Our results show that the Gemma 7B embeddings yield the best regression performance, leading us to use Gemma 7B in subsequent experiments.

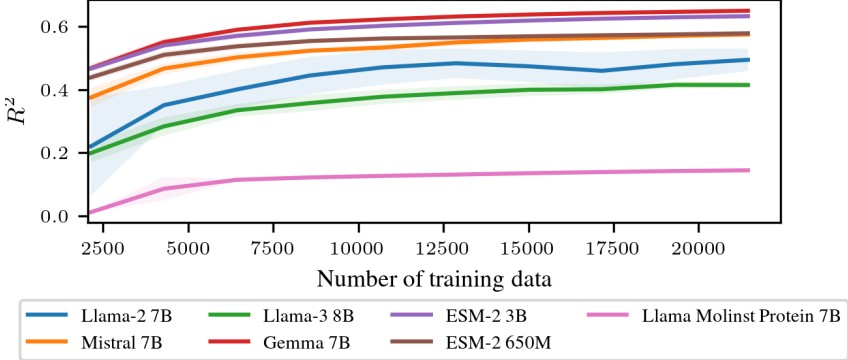

Figure 17: Coefficient of determination on the test set as a function of the number of training data. The $R^2$ metric is used to evaluate the performance of embedding protein sequences.

## G.2    MODELING PROTEIN SEQUENCE DESIGN WITH NATURAL LANGUAGE

We employ the instruction-finetuned Llama-3.2 3B model as our variational network. The model is available online at `https://huggingface.co/meta-llama/Llama-3.2-3B-Instruct`. We frame the protein design process as a dialogue to leverage the model's conversational capabilities. Specifically, we prompt the model to generate the next protein sequences based on previously observed protein and their fluorescent level. The prompts we used are outlined below.

System prompt:

```
You are a helpful assistant who works in a protein engineering lab. We are trying to edit
    ↪ a given protein by a sequence of 1-step protein editing, known as mutation. You
    ↪ need to use your knowledge to help me propose suitable protein editing. Going from
    ↪  an initial protein to an optimal one can take many steps.
```

First prompt:

```
Edit 1 amino acid in the below protein sequence to create a new protein with higher
    ↪ fluorescence. The amino acid must be in set {D, E}. Protein sequence: {
    ↪ starting_protein}
```

**Feedback prompt:**

```
Fluorescence level of the above protein: {fluorescence_level} Based on the above protein
    ↪ sequence and its fluorescence value, edit 1 amino acid to achieve higher
    ↪ fluorescence. You must only return the modified protein sequence and nothing else.
    ↪  Modified protein sequence:
```

## G.3 SUPERVISED FINE-TUNING PROCESS

Before starting the BO process, we conduct supervised fine-tuning (SFT) on the variational network to adapt it to the protein design task. We generate a dataset for SFT training consisting of 100 dialogues, each containing $L$ rounds corresponding to the number of lookahead steps. The proteins in each dialogue are created by either randomly mutating or retaining the previous protein. The fine-tuning hyperparameters are provided below.

Table 4: SFT hyperparameters.

| Hyperparameter | Value |
|---|---|
| Learning rate | $10^{-4}$ |
| Epochs | 3.3 |
| Batch size | 4 |
| Learning rate warmup ratio | 0.1 |
| Learning rate schedule | Cosine |
| LoRA $\alpha$ | 32 |
| LoRA $r$ | 16 |
| LoRA dropout | 0.1 |
| LoRA target modules | q_proj, v_proj |

## G.4 NONMYOPIC BAYESIAN OPTIMIZATION AS MULTI-TURN PROXIMAL POLICY OPTIMIZATION

Proximal Policy Optimization (PPO) (Schulman et al., 2017) is typically used to fine-tune language models for single-turn conversations, where the model responds once to a prompt without considering future turns. However, our approach requires the model to think ahead and generate multiple future queries (in this case, protein sequences) over several turns. To address this, we modify existing PPO frameworks to handle multiturn conversations, allowing the model to generate and optimize future sequences during training. We also use vLLM (Kwon et al., 2023), a system designed to improve the speed and efficiency of inference (i.e., generating outputs from the model). However, vLLM is an inference engine and cannot be used directly for training (Kwon et al., 2023). To overcome this, after each step of updating the model during training (called a gradient step), we transfer the updated model's weights (parameters) to the vLLM system. This allows us to use vLLM for faster generation of outputs, leading to more efficient training. Additionally, running vLLM with PPO simultaneously on multiple GPUs is challenging for large language models due to differences in how vLLM and PPO handle GPU VRAM. To address this, we employ Ray Moritz et al. (2018) to isolate the GPU environments for vLLM and PPO, creating a pipeline that enables efficient weight synchronization between them. Figure 18 illustrates the process of weight syncing between PPO and vLLM.

In the PPO training process, we calculate a final reward for each dialogue using a function $\ell$. This function varies depending on the acquisition method being used (e.g., expected improvement or simple regret). Once the reward is computed, it is adjusted, or "discounted," for each individual turn in the dialogue. This means that actions taken earlier in the conversation get less reward compared to later actions. We then use this discounted reward as feedback to update the model during PPO training. By doing this, we extend the single-turn PPO framework, which normally handles one response at a time, to work for our multiturn conversation data. The hyperparameters used for fine-tuning PPO are provided below.

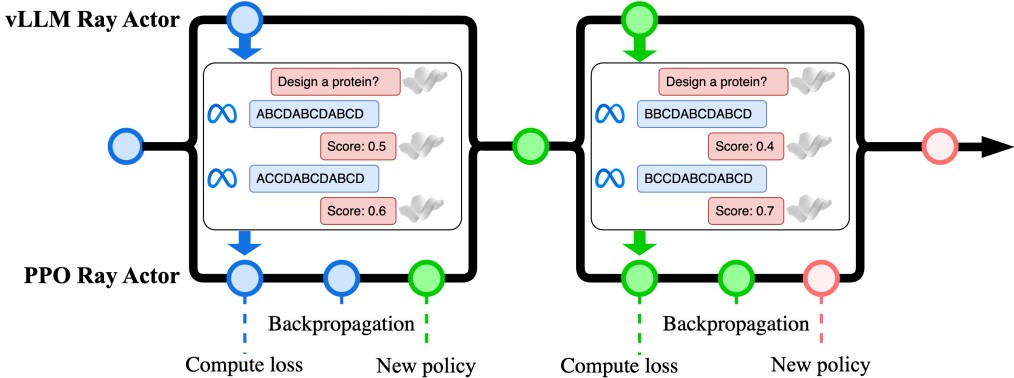

Figure 18: The designed PPO pipeline with vLLM using Ray involves the vLLM actor generating multi-turn protein refinements and calling a surrogate model to compute reward scores. These scores, along with the generated data, are passed to the PPO actor to compute loss, backpropagate, and update the LLM weights. The updated weights from the PPO actor are then synced to the vLLM actor for the next generation.

Table 5: PPO and rollout hyperparameters.

| Hyperparameter | Value |
|---|---|
| Learning rate | $10^{-4}$ |
| Epochs | 64 |
| Batch size | 1 |
| Learning rate warmup ratio | 0.1 |
| Learning rate schedule | Cosine |
| LoRA $\alpha$ | 256 |
| LoRA $r$ | 128 |
| LoRA dropout | 0.1 |
| LoRA target modules | q_proj, v_proj |
| Maximal rollout retry | 32 |
| Discount reward factor | 0.95 |

## H ABLATION STUDY ON PROTEIN DESIGN EXPERIMENTS

We conduct an ablation study using two different starting proteins and two distinct synthetic functions $g(x)$ to construct diverse protein spaces, enabling a systematic analysis of their characteristics. The two synthetic functions are defined as follows:

$$\begin{aligned} g_1(x) &= -0.005(d - 0.5)(d - 5)(d - 8)(d - 13.4), \\ g_2(x) &= -e^{-0.7 \cdot \sqrt{0.5 \cdot d^2}} - e^{0.5 \cdot \cos(0.4\pi d)} + e + 0.3. \end{aligned} \quad (2)$$

Here, $g_1(x)$ represents a polynomial function that introduces one local maxima across the input space, while $g_2(x)$ is a more complex one with two local maxima. Visualizations of the resulting protein spaces, derived from the combination of starting proteins and synthetic functions, are shown in Figure 19.

We present the results of additional experiments on protein design with the same starting protein with $g_2$ (Figure 20 top), and with a different starting protein with $g_1$ (Figure 20 bottom). These figures demonstrate that our proposed nonmyopic method outperforms other myopic baselines in various settings regardless of different starting proteins or synthetic value functions.

We visualize the designed proteins in the experiments of starting protein #1 and $g_1$. We use ESM-Fold (Lin et al., 2022) to fold the designed proteins and PyMol (Schrödinger, LLC, 2015) to visualize them. The visualizations are presented in Table 7.

Table 6: Protein space constraints

| No. | Starting protein | Allowed positions | Allowed AAs |
|---|---|---|---|
| #1 | SKGEELFTGVVPILVELGGDVNGHKFSVSGEGEGDAT
YGKLTLKFICTTGKLPVPWPTLVTTLSYGVQCFSRFP
DHMKQHDFFKSAMPEGYVQERTIFSKDDGNYKTRAEV
KFEGDELVNRIELKGIDFKEEENILGHKLEENYNSHN
VYIMADDQKNGIKVNFKIRHNIEDDSVQLADHYQQNT
PIGDEPVLLPDDHYLSTQSALSKDDNEDRDEMVLLEF
VTAAGITHGMDELYK | 116, 131, 132, 141,
154, 171, 172, 189,
196, 209, 212, 215 | E, D |
| #2 | SKPEELFTPVVGILVELDPDVNGHKFSVSGEGEPDAT
YGKLTLKFICTTGKLGVGWGTLVTTLSYGVQCFSRYP
DHMKQHDFFKSAMPEGYVQERTIFFKDDGNYKTRAEV
KFEPDTLVNRIELKGIVFKEDGNTLGHKLEYNYNSHN
VYIMADEQKNGIKVNFKIRHNIEDGSVQLADHYQQNT
PIPDGPVLLPDNHYLSTQSALSKDPNEKRDHMVLLEF
VTAAGITHGMDELYK | 2, 8, 11, 18, 33,
52, 54, 56, 114, 158,
187, 190 | G, P |

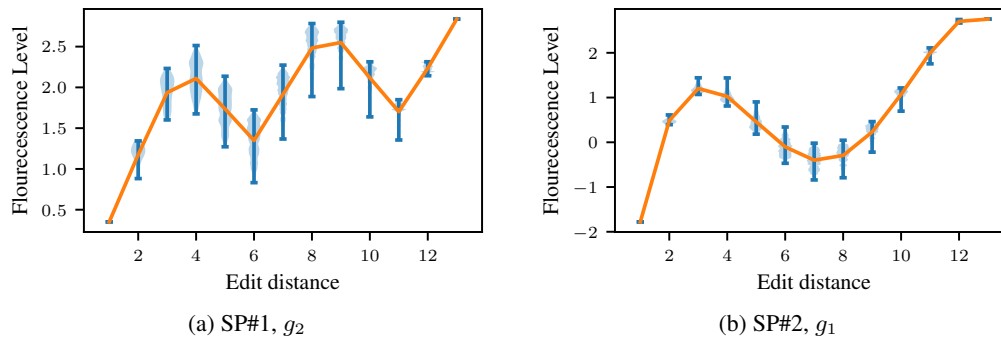

(a) SP#1, $g_2$        (b) SP#2, $g_1$

Figure 19: Ablation of protein spaces with varying starting proteins and synthetic functions

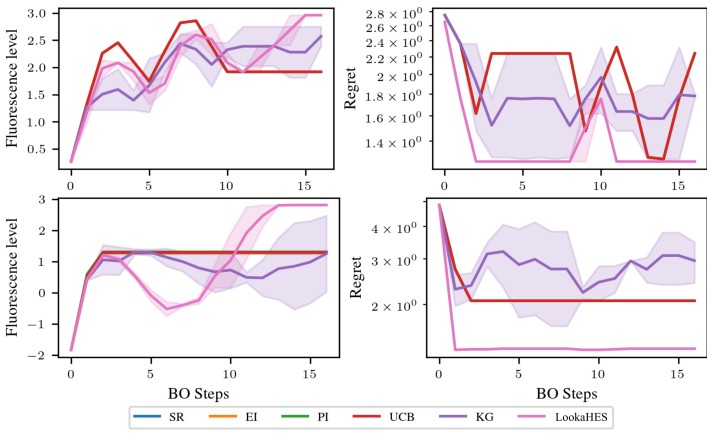

Figure 20: Fluorescence levels (left) and regret (right) observed over Bayesian Optimization (BO) steps. The top row shows results for experiments starting with protein #1 and function $g_2$, while the bottom row corresponds to experiments starting with protein #2 and function $g_1$.

Table 7: Visualization of designed proteins. Edited amino acids are highlighted in red.

| Sequence | 3D Structure |
|---|---|
| **Starting protein:** SKGEELFTGVVPILVELGGDVNGHKFSVSG EGEGDATYGKLTLKFICTTGKLPVPWPTLVTTLSYGVQCFSR FPDHMKQHDFFKSAMPEGYVQERTIFSKDDGNYKTRAEVKFE GDELVNRIELKGIDFKEEENILGHKLEENYNSHNVYIMADDQ KNGIKVNFKIRHNIEDDSVQLADHYQQNTPIGDEPVLLPDDH YLSTQSALSKDDNEDRDEMVLLEFVTAAGITHGMDELYK | 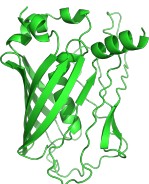 |
| **LookaHES - Optimal:** SKGEELFTGVVPILVELGGDVNGHKF SVSGEGEGDATYGKLTLKFICTTGKLPVPWPTLVTTLSYGVQ CFSRFPDHMKQHDFFKSAMPEGYVQERTIFSKDDGNYKTRAE VKFEGD**D**LVNRIELKGIDFKE**DD**NILGHKLE**D**NYNSHNVYIM AD**E**QKNGIKVNFKIRHNIE**EE**SVQLADHYQQNTPIGD**D**PVLL PD**E**HYLSTQSALSKD**E**NE**E**RD**D**MVLLEFVTAAGITHGMDELY K | 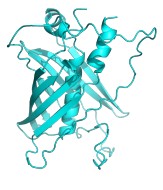 |
| **SR:** SKGEELFTGVVPILVELGGDVNGHKFSVSGEGEGDATYG KLTLKFICTTGKLPVPWPTLVTTLSYGVQCFSRFPDHMKQHD FFKSAMPEGYVQERTIFSKDDGNYKTRAEVKFEGDELVNRIE LKGIDFKEE**D**NILGHKLEENYNSHNVYIMADDQKNGIKVNFK IRHNIEDDSVQLADHYQQNTPIGD**D**PVLLPDDHYLSTQSALS KDDNEDRDEMVLLEFVTAAGITHGMDELYK | 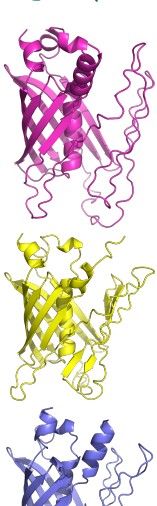 |
| **EI, PI, UCB:** SKGEELFTGVVPILVELGGDVNGHKFSVSGEGE GDATYGKLTLKFICTTGKLPVPWPTLVTTLSYGVQCFSRFPD HMKQHDFFKSAMPEGYVQERTIFSKDDGNYKTRAEVKFEGDE LVNRIELKGIDFKE**D**ENILGHKLEENYNSHNVYIMADDQKNG IKVNFKIRHNIEDDSVQLADHYQQNTPIGDEPVLLPDDHYLS TQSALSKD**E**NEDRDEMVLLEFVTAAGITHGMDELYK | |
| **KG:** SKGEELFTGVVPILVELGGDVNGHKFSVSGEGEGDATYG KLTLKFICTTGKLPVPWPTLVTTLSYGVQCFSRFPDHMKQHD FFKSAMPEGYVQERTIFSKDDGNYKTRAEVKFEGDELVNRIE LKGIDFKEEENILGHKLEENYNSHNVYIMADDQKNGIKVNFK IRHNIEDDSVQLADHYQQNTPIGDEPVLLPDDHYLSTQSALS KDDNE**E**RDEMVLLEFVTAAGITHGMDELYK | |

