# OpenReview forum: "Neural Nonmyopic Bayesian Optimization in Dynamic Cost Settings"
_ICLR.cc/2025/Workshop/AgenticAI — ICLR 2025 Workshop AgenticAI Oral_

### Official Review · Reviewer_tiGq · 2025-03-03
**Meaningful problem setting with solution of limited innovation**

**Rating:** 5
**Confidence:** 4

**Review:**

This paper proposes a nonmyopic Bayesian optimization method for black-box optimization with action-dependent evaluation costs. It optimizes recurrent neural networks for sampling and uses pathwise sampling to reduce complexity. Experiments on synthetic functions and protein design validate its effectiveness and practicality.

Strengths:
1. This paper proposes a model to address the problem of black box optimization problem with dynamic cost. Some practical examples of this problem are provided to support the significance and impact of the problem setting.

2. This paper conducts experiments on both synthetic continuous data and real-world discrete data (protein optimization), to show the effectiveness of the proposed model.

Weaknesses:
1. The contribution of this paper in the aspect of machine learning innovation is limited. This paper addresses the key point of its research question, i.e., the dynamic cost, by considering the history moves as a sequence and using RNN to capture the trend. Besides this, so far as I can see, this paper does not provide innovative techniques that consider the mapping between the moves and costs.

2. This paper does not provide convincing enough empirical analysis. The baselines chosen are mostly before 2010, while there are other  available state-of-the-art models that are related with this paper, such as the following one (also cited by this paper):

Lee, Eric Hans, et al. "A nonmyopic approach to cost-constrained Bayesian optimization." Uncertainty in Artificial Intelligence. PMLR, 2021.

The above work is highly related to this paper but is not compared through experiments.

---

### Official Review · Reviewer_uG3H · 2025-03-03

**Rating:** 7
**Confidence:** 3

**Review:**

This paper introduces LookaHES, a novel nonmyopic Bayesian optimization algorithm designed for dynamic cost settings where query costs depend on previous actions. LookaHES integrates a neural network policy to achieve scalability in planning multiple steps ahead. The authors benchmark LookaHES against several baseline methods on nine synthetic functions with various dimensions and noise levels, NASA satellite images, and on a real-world protein sequence design problem. The results demonstrate superior performance over myopic baselines and scalability to 20-step lookahead horizons.

Strengths:

1.	The paper addresses an important problem of nonmyopic Bayesian optimization with dynamic costs, addressing a gap in existing literature that primarily focuses on static cost structures.

2.	As a solution to the research question, the authors propose LookaHES. By using a neural network policy for variational optimization, LookaHES can handle lookahead horizons of up to 20 steps, significantly beyond state-of-the-art nonmyopic approaches.

3.	Empirical evaluations across noise levels, cost structures, and dimensions demonstrate the efficiency and solution quality of proposed LookaHES method. The paper also contains results for the protein sequence design, showing how the method can be applied to real-world problems.

Concerns:

1. Although this paper compares against several baseline methods, all except MSL are myopic methods, if I understand correctly. Conducting additional comparisons with state-of-the-art nonmyopic methods would strengthen the evaluation.

2. LookaHES focuses on a known dynamic cost structure, which may not always be feasible. The paper does not address scenarios where costs are uncertain with high variability.

3. The performance of LookaHES relies on a well-specified surrogate model, i.e., the surrogate model should approximate the target function effectively. As shown in Figure 12, using Bayesian linear regression resulted in a wrong approximation of the ground truth functions. I am curious: how does model misspecification influence planning quality, and would it be a limitation for applying the method to complex real-world decision-making scenarios?

---

### Official Review · Reviewer_oKCy · 2025-03-05
**Good Paper**

**Rating:** 7
**Confidence:** 4

**Review:**

The paper introduces LookaHES, a neural nonmyopic Bayesian optimization algorithm designed to optimize black-box functions in dynamic cost environments. Traditional Bayesian optimization (BO) methods assume static query costs, which limits their applicability to real-world problems where costs change based on prior queries, such as geological surveys and biological sequence design. LookaHES addresses this by incorporating a neural network policy that enables multi-step lookahead while considering dynamic costs, making strategic investments in queries that may initially seem suboptimal but unlock better solutions over time. The method is evaluated on synthetic functions, NASA satellite imagery, and protein sequence design, demonstrating superior efficiency and solution quality compared to traditional myopic approaches. LookaHES outperforms existing BO algorithms by efficiently handling large, complex search spaces and optimizing queries over longer planning horizons.

Suggested Improvements:
1. The authors should explain how the neural network learns to balance cost and reward in different environments.
2. Test LookaHES on real-world engineering or logistics problems, such as robotic path planning or drug discovery.

---

### Decision · Program_Chairs · 2025-03-05

Accept (Oral)